# Fabrication of Encapsulated Gemini Surfactants

**DOI:** 10.3390/molecules27196664

**Published:** 2022-10-07

**Authors:** Bogumił Brycki, Adrianna Szulc, Iwona Kowalczyk, Justyna Brycka

**Affiliations:** 1Department of Bioactive Products, Faculty of Chemistry, Adam Mickiewicz University Poznan, Uniwersytetu Poznanskiego 8, 61–614 Poznan, Poland; 2MDA Sp. z o.o., Wolczynska 18, 60-003 Poznan, Poland

**Keywords:** gemini surfactants, encapsulation, alginate capsules, gelatin microsphere

## Abstract

(1) Background: Encapsulation of surfactants is an innovative approach that allows not only protection of the active substance, but also its controlled and gradual release. This is primarily used to protect metallic surfaces against corrosion or to create biologically active surfaces. Gemini surfactants are known for their excellent anticorrosion, antimicrobial and surface properties; (2) Methods: In this study, we present an efficient methods of preparation of encapsulated gemini surfactants in form of alginate and gelatin capsules; (3) Results: The analysis of infrared spectra and images of the scanning electron microscope confirm the effectiveness of encapsulation; (4) Conclusions: Gemini surfactants in encapsulated form are promising candidates for corrosion inhibitors and antimicrobials with the possibility of protecting the active substance against environmental factors and the possibility of controlled outflow.

## 1. Introduction

Surfactants are chemical compounds that upon adsorption at the air–water interface reduce the surface tension of water. Their molecules contain at least two moieties, hydrophobic and hydrophilic. The hydrophobic part is usually a straight or branched hydrocarbon or fluorocarbon chain with 8–18 carbon atoms, whereas a hydrophilic moiety is a polar or ionic group. The head group can be cationic, anionic, amphoteric or nonionic [1]. The balance between hydrophobic and hydrophilic parts, and the hydrophilic-lipophilic balance, is responsible for the self-assembly process of these amphiphilic compounds in solutions [2]. It affects the properties of the compounds, which generates their use in many areas of life and industry, starting from cleaning agents and detergents [3,4,5,6], through to protective applications such as antimicrobials [7,8,9,10] or anticorrosion products [11,12,13,14,15], to modern biomedical applications [16,17,18,19,20,21] or enhanced oil recovery [22,23,24,25,26]. Surfactants may also have two hydrocarbon chains attached to a polar head and are named double chain surfactants. They can be also anionic, cationic, amphoteric or nonionic, but of greatest practical importance are their double quaternary ammonium salts, named as gemini surfactants by Menger and Littau [27]. Cationic gemini surfactants are made of two single alkylammonium monomeric salts connected by a spacer. The nature of the spacer can be very different; it can be a flexible polymethylene chain or a rigid aromatic ring, or a short or long hydrocarbon chain. Spacers can be hydrophobic or hydrophilic in nature with additional polar organic groups [2,28,29,30,31]. The type of spacer determines the properties of the gemini surfactants, significantly influencing their hydrophilic-lipophilic balance and potential application [32,33,34,35,36]. The substituents in gemini surfactants are also responsible to a great extent for the behavior of these compounds in solution and their possible applications [2,37]. From an ecological point of view, the functionalization of substituents with groups of natural origin is of great importance, because it can increase their biological profile, improving biodegradability or reducing ecotoxicity [8,38]. Gemini surfactants are referred to as multifunctional, because one compound may be an exceptionally effective surface active agent [39,40,41], corrosion and biocorrosion inhibitor [42,43,44,45,46,47], but also has remarkably high antibacterial [48,49,50,51] and antifungal [52,53,54,55,56,57] activity and is effective in preventing biofilm eradication [52,58]. It is worth emphasizing that gemini surfactants work effectively at a very low concentration, often several times smaller in comparison to single quaternary ammonium salts [59,60,61,62,63]. This is very important from an ecological point of view, in that to achieve the desired utility effect, the minimal concentration of compounds are used [64,65,66,67]. 

Encapsulation is a process of enclosing or entrapping a core material (active agent) inside a solid shell or within a solid or liquid matrix. The different encapsulation techniques can provide various microcapsule morphologies (Figure 1) [68,69].

Recent decades have witnessed the rapid development of encapsulation techniques leading to the industrial use of capsules, especially in food and pharmaceutical/medicinal industries. Encapsulation in the food industry ensures several functions, including masking of undesirable color/flavor/taste, preservation of unstable components, incorporation of additional functional and nutritional constituents and site-specific release of encapsulated ingredients at a controlled time and rate [70,71,72]. In the pharmaceutical industry, encapsulation is commonly adopted in drug delivery systems to form capsules that improve functionality and solubility in order to protect the active agent of the medicine and prevent it from leaching out before reaching the targeted site [73,74,75,76]. Generally, the main purpose of the encapsulation is to achieve a surface with a permanent acting active agent or a surface with a slowed and, in many cases, controlled release of the substance. The release can be spontaneous or triggered by some kind of stimuli, such as humidity, pH, temperature, light, magnetic field, CO_2_ (acidification), redox and enzymes or mechanical impact [69]. The immobilization of biocidal agents by the encapsulation technique is another way to overcome some problems associated with the toxicity and the lifetime of conventional strategies for the incorporation of agents. For many substances, e.g., antimicrobials are chemically fixed in a microcapsule shell where the uncontrolled release is strongly restricted. Therefore, encapsulated microbiocides are potentially non-toxic and environmentally friendly materials [77,78,79]. Recently, encapsulated corrosion inhibitors have started to be put into use to ensure constant access of the active substance to protect the material according to slow and controlled release or to fill the surface with self-healing properties [80,81,82,83,84,85]. Very popular carriers in encapsulation processes are natural polymers, especially proteins and polysaccharides including agarose, alginate, carrageenan, chitosan, hyaluronic acid, collagen, elastin, gelatin, fibrin, and silk fibroin [86]. Thus, it is essential to select the appropriate biopolymer material to formulate nanoparticles for practice applications. Polysaccharides have attracted considerable interest in this area due to their good biocompatibility and biodegradability and low costs [87]. 

The purpose of this paper is to report the fabrication of biodegradable encapsulated gemini surfactants. In reference to the need to discover materials with good biological profiles, we used gelatin and sodium alginate as a shell materials and gemini surfactants functionalized by ester groups in substituents parts. These kinds of gemini surfactants exhibit great antimicrobial [88,89,90] or anticorrosion [91] properties and better biodegradability than conventional analogues [36,92,93,94]. To our knowledge, the encapsulation technique using alginate [95,96,97] and gelatin [98,99] is well-known and described, but information about encapsulated gemini surfactants is lacking.

## 2. Results and Discussion

### 2.1. Synthesis

Gemini surfactants (GS) were obtained by innovative synthesis without a solvent (Figure 1), according to the preparation developed in our laboratory and described previously [100]. This method ensures the best results in the shortest period of time. The reactions proceed according to the mechanism of nucleophilic substitution S_N_2, which go most quickly when the concentrations of the reagents are the highest. This is what makes synthesis possible without a solvent. Moreover, such reactions are in line with the assumptions of green chemistry, as the use of reagents and energy is limited. The synthetic details are given in the Appendix A.

In the presented syntheses, we obtained six products, four gemini surfactants with ester bonds in substituents. For comparison purposes, classic gemini surfactants with a hydrocarbon chain in the substituent were also prepared. Particularly noteworthy is the compound 3-oxa-1,5-pentane-bis(*N*-dodecyl-*N,N*-dimethylammonium bromide) (12-O-12), which exhibits the unique antimicrobial and surface activity described by us earlier [29,67]. The structural formulas and designations of the obtained products are shown in Figure 2. 

### 2.2. Analysis of Gemini Surfactants

The purities of synthesized gemini surfactants were also confirmed by spectroscopy methods (^1^H NMR, ^13^C NMR, FTIR).

10(E)-3-10(E) Cl: mp 155–156 °C; ^1^H NMR (403 MHz, CDCl_3_) δ 4.85 (4H, N^+^C**H**_2_COO-), 4.17 (4H, O-C**H**_2_-), 4.00 (4H, N^+^C**H**_2_- spacer), 3.65 (12H, N^+^C**H**_3_), 2.72 (2H, -C**H**_2_- spacer), 1.66 (4H, O-CH_2_-C**H**_2_-), 1.30-1.26 (28H, O-CH_2_-CH_2_-(C**H**_2_)_7_-), 0.88 (6H, -CH_2_C**H**_3_); ^13^C NMR (101MHz, CDCl_3_) δ 164.90 (**C**=O), 66.87 (N^+^**C**H_2_COO-), 61.76 (O-**C**H_2_-), 61.41(N^+^**C**H_2_- spacer), 52.05 (N^+^**C**H_3_), 29.53, 29.47, 29.28, 29.19, 28.24, 22.65 (O-CH_2_-(CH_2_)_8_-), 25.61 (-CH_2_- spacer), 14.09 (-CH_2_CH_3_).

10(E)-O-10(E) Cl: mp 149–151 °C; ^1^H NMR (403 MHz, CDCl_3_) δ 4.97 (4H, N^+^C**H**_2_COO-), 4.33 (8H -C**H**_2_- spacer), 4.16 (4H, O-C**H**_2_-), 3.71 (12H, N^+^C**H**_3_), 1.65 (4H, O-CH_2_-C**H**_2_-), 1.31–1.25 (28H, O-CH_2_-CH_2_-(C**H**_2_)_7_-), 0.88 (6H, -CH_2_C**H**_3_); ^13^C NMR (101MHz, CDCl_3_) δ 165.06 (**C**=O), 66.71 (N^+^**C**H_2_COO-), 64.54 (-**C**H_2_O spacer), 62.03 (N^+^**C**H_2_ spacer), 61.36 (O-**C**H_2_-), 52.41 (N^+^**C**H_3_), 29.52, 29.45, 29.26, 29.17,28.27, 22.62 (O-CH_2_-(**C**H_2_)_8_-), 14.08 (-CH_2_**C**H_3_). 

12(E)-O-12(E) Cl: mp 162–163 °C; ^1^H NMR (403 MHz, CDCl_3_) δ 4.92 (4H, N^+^C**H**_2_COO-), 4.16 (8H -C**H**_2_- spacer), 3.75 (4H, O-C**H**_2_-), 3.69 (12H, N^+^C**H**_3_), 1.70 (4H, O-CH_2_-C**H**_2_-), 1.39–1.25 (36H, O-CH_2_-CH_2_-(C**H**_2_)_9_-), 0.87 (6H, -CH_2_C**H**_3_); ^13^C NMR (101MHz, CDCl_3_) δ 165.05 (**C**=O), 66.75 (N^+^**C**H_2_COO-), 64.43 (-**C**H_2_O spacer), 62.15 (N^+^**C**H_2_ spacer), 59.00 (O-**C**H_2_-), 52.59 (N^+^**C**H_3_), 31.82, 29.47, 29.42, 29.16, 28.22, 25.61, 22.64 (O-CH_2_-(**C**H_2_)_10_-), 14.10 (-CH_2_**C**H_3_). 

12(E)-O-12(E) Br: mp 171–172 °C; ^1^H NMR (403 MHz, CDCl_3_) δ 4.95 (4H, N^+^C**H**_2_COO-), 4.37 (8H -C**H**_2_- spacer), 4.17 (4H, O-C**H**_2_-), 3.74 (12H, N^+^C**H**_3_), 1.66 (4H, O-CH_2_-C**H**_2_-), 1.30–1.26 (36H, O-CH_2_-CH_2_-(C**H**_2_)_9_-), 0.88 (6H, -CH_2_C**H**_3_); ^13^C NMR (101MHz, CDCl_3_) δ 164.84 (**C**=O), 66.84 (N^+^**C**H_2_COO-), 64.43 (-**C**H_2_O spacer), 62.57 (N^+^**C**H_2_ spacer), 62.15 (O-**C**H_2_-), 52.68 (N^+^**C**H_3_), 31.88, 29.61, 29.32, 29.17, 28.27, 25.62, 22.66 (O-CH_2_-(**C**H_2_)_10_-), 14.08 (-CH_2_**C**H_3_). 

12-3-12: mp 213–215 °C; ^1^H NMR (403 MHz, CDCl_3_) δ 3.88 (4H, N^+^C**H**_2_-), 3.54 (4H, N^+^C**H**_2_CH_2_-), 3.42 (12H, N^+^C**H**_3_), 2.74 (2H -C**H**_2_- spacer), 1.80 (4H, N^+^-CH_2_-C**H**_2_-), 1.36–1.26 (36H, N^+^-CH_2_-CH_2_-(C**H**_2_)_9_-), 0.88 (6H, -CH_2_C**H**_3_); ^13^C NMR (101MHz, CDCl_3_) δ 66.51 (N^+^**C**H_2_-), 60.92 (N^+^-**C**H_2_- spacer), 51.16 (N^+^**C**H_3_), 31.81, 29.53, 29.42, 29.37, 29.25, 29.18, 26.88 (N^+^-CH_2_-(**C**H_2_)_10_-), 22.60 (-**C**H_2_- spacer), 14.05 (-CH_2_**C**H_3_). 

12-O-12: mp 248–249 °C; ^1^H NMR (403 MHz, CDCl3) δ 4.35 (4H,-OC**H**_2_ spacer), 4.04 (4H, N+C**H**_2_CH_2_O- spacer), 3.63 (4H, N^+^C**H**_2_-) 3.46 (12H, N^+^C**H**_3_), 1.73 (4H, N^+^CH_2_C**H**_2_-), 1.45–1.17 (36H, N^+^-CH_2_-CH_2_-(C**H**_2_)_9_-) -), 0.88 (6H, -CH_2_C**H**_3_); ^13^C NMR (101MHz, CDCl_3_) δ 65.81 (N^+^**C**H_2_CH_2_O spacer), 64.57 (-O**C**H_2_ spacer), 63.94 (N^+^**C**H_2_-), 51.60 (N^+^**C**H_3_), 31.82, 29.53, 29.46, 29.38, 29.28, 29.26, 26.27,22.60 (N^+^-CH_2_-(**C**H_2_)_10_-), 14.05 (-CH_2_**C**H_3_).

The Fourier transform infrared spectroscopy (FTIR) spectra for all obtained gemini surfactants have typical bands for these kind of chemical compounds (Table 1).

### 2.3. Fabrication and Analysis of Alginate Capsules (Al@GS)

In encapsulation of gemini surfactants by sodium alginate, we used complex formation, which can occur in an aqueous solution. We used calcium chloride as a crosslinker. Firstly, in a clear aqueous solution of gemini surfactant, sodium alginate was dissolved. The mixture was heated and stirred until homogeneous. It was then injected into a cross-linking solution containing calcium chloride. Sodium alginate is a polyelectrolyte biopolymer extracted from brown algae. It can be characterized as an anionic copolymer, which consists of 1–4 linked α-L-guluronic and β-D-mannuronic residues with a very diverse composition and structure sequence. Alginates are considered block copolymers. Alginates form a gel in the presence of divalent cations such as Ca^2+^. After interaction with the ions of divalent metals, they form an ordered structure. These cations act as cross linkers between the functional groups of the alginate chains. Calcium alginate gelation is an irreversible and almost immediate process. This process is governed by the relative rate of diffusion of calcium ions and polymer molecules into the gelling zone and can therefore be expressed by the relationships used for other diffusion limited reaction systems [101]. After filtration from the aqueous solution, the encapsulated gemini surfactants were dried in an incubator, thus reducing their diameter several times, and reducing the weight by over 95% (Figure 3). The encapsulation efficiency was calculated as the ratio of the amount of encapsulated to the amount of gemini surfactant used, and in each case it exceeded 90%.

Initial analysis of the capsule surface morphology was performed using a digital microscope. In Figure 3, we presented capsules of gemini surfactant 12-O-12. Subpoint (a) and (b) present ones with the addition of barium sulphate, the addition of which strengthens the coating and maintains its shape after drying. Subpoint (c) and (d) show capsules obtained without the addition of barium sulphate, which are transparent and visibly more deformed when dried. In order to compare the sizes, wet (a) and (c) and dry (b) and (d) capsules were put together. The addition of barium sulphate improves the strength of the capsules and also gives them a more uniform shape. However, it does not affect the size of the capsules; the average diameter of wet microcapsules is 10 mm a dry is 1.5 mm. The structure of the gemini surfactant used does not affect the size or appearance of the alginate capsules. Introduction of hydrophilic ester groups into gemini surfactants substituents has no effect on preparation and morphology of both wet and dry capsules (Figure 4).

Detailed studies of the surface morphology of alginate capsules were performed using a scanning electron microscope. By comparing the appearance and morphology of the surface of the empty capsules, it can be confirmed that the addition of barium sulphate results in the strengthening of the shell. It is visible in the macroscopic image as a white color. Moreover, alginate capsules prepared with BaSO_4_ are characterized by a more symmetrical, spherical structure (Figure 5).

The alginate capsules filled with gemini surfactants apparently have a different, more varied morphology than the empty capsules (Figure 6). This proves that they contain an active substance. It can be stated that in the SEM images, the empty capsules are smoother, whereas those containing gemini surfactants are more pleated. The addition of barium sulphate in the case of capsules filled with gemini surfactant 12-O-12 (Al@12-O-12) strengthens the shell. The surface morphology of the capsules does not depend on the structure of the gemini surfactant (Figure 7).

In order to confirm the encapsulation, FTIR spectra (ATR technic) were made after the capsules were thoroughly washed with water and dried. In Figure 8a**,** there is a spectrum of empty alginate capsules (Al@empty). The most intense band in the spectrum at 3347 cm^−1^ comes from the O-H stretching vibrations of the O-H hydroxyl group. The broadening of the band indicates the presence of hydrogen bonds. Another characteristic band at 1598 cm^−1^ is derived from the carbonyl group of the carboxylate. Remaining spectra derived from capsules filled with gemini surfactants prove the correct encapsulation path. In Figure 8b, there is a spectrum of alginate capsules with gemini surfactant 12-O-12 (Al@12-O-12) and in addition to the characteristic bands of alginate, bands derived from gemini surfactant can be seen, especially the bands coming from the C-H stretching vibrations of the methylene and methyl groups at 2854 cm^−1^ and 2922 cm^−1^. The last spectrum from capsules filled with gemini surfactant with ester groups (Al@10(E)-O-10(E) Cl) looks analogous, which has an additional band derived from ester carbonyl at 1748 cm^−1^.

### 2.4. Fabrication and Analysis of Gelatine Capsules (Ge@GS)

Gelatin is a protein derived by hydrolysis from collagen, a naturally occurring protein. In the formation of gels, particles and microspheres gelatin is used particularly frequently in the pharmaceutical industry due to biocompatibility and non-toxicity [99]. Cross-linking of gelatin fibers can be carried out by various chemical methods (using sugars, dialdehydes, phenolic compounds), enzymatic and physical means or a combination of these [102]. In our work, we carried out thermal cross-linking with glucose. In this reaction mechanism, the aldehyde group of reducing sugars can react with the free amino groups of gelatin molecule resulting in the formation of an aminoglycoside. The aminoglycoside can further react with another amine group creating a cross-linked structure [99]. A water-in-oil emulsion was used. The dispersion medium was sunflower oil, and the surfactant used was Span 80 (sorbitan oleate), which was used to minimize aggregation of gelatin microcapsules in the oil solution. After filtration, the gelatin capsules with gemini surfactants were dried in an er, thus the desiccator reduced their diameter several times, and reduced the weight by over 95% (Figure 9).

Each capsule was characterized by a compact and wrinkled consistency, and on its surface one could see gelatin microcapsules stuck together. The most durable and, at the same time, the hardest form were empty gelatin capsules, while those filled with gemini surfactants were softer. Additionally, capsules with gemini surfactants showed less aggregation. All of the obtained capsules turned white. The encapsulation efficiency was calculated as the ratio of the amount of encapsulated to the amount of gemini surfactant used and, in each case, it exceeded 60%. The surface morphology of the gelatin capsules was analyzed on the basis of SEM images. Figure 10 presents a comparison of gelatin microspheres with and without gemini surfactant. The structure of gemini surfactant has an influence on morphology of gelatin microspheres (Figure 11).

SEM analysis of gelatin capsules filled with gemini surfactants have a very symmetrical and spherical structure in comparison to empty capsules. This suggests that gemini surfactants are effective in stabilizing gelatin capsules, which is particularly evident in the case of compound 12(E)-O-12(E) Br. The type of active substance used also has a great influence on the size of the capsules. They have an average diameter 12.5 µm and 20 µm for Ge@12(E)-O-12(E) Cl and Ge@12(E)-O-12(E) Br, respectively.

The FTIR spectra of empty capsules and capsules filled with gemini surfactant confirm the effectiveness of the encapsulation preparation (Figure 12).

In Figure 12a, it can be seen that the band in the area of 3700–3300 cm^−1^ is due to the N-H stretching of amine and the amine group, which is further widened by the presence of hydrogen bonds. Bands at 1631 and 1515 cm^−1^ come from stretching vibrations of carbonyl and C-N bonds. In the second spectrum, separate from the bands characteristic of the cross-linked gelatin, bands originating from gemini surfactant can be found.

## 3. Materials and Methods

### 3.1. Materials

1-Decanol (98%) and 1-dodecanol (98%) were obtained from Merck KGaA (Darmstadt, Germany). Chloroacetic acid (99%), bromoacetic acid (97%), 1-bromododecane (97%), *N,N,N**′,N*′-tetramethyl-1,3-propanediamine (TMPDA) (99%), bis[2 -(*N,N*-dimethylamino)ethyl] ether (BDMAEE) (97%), D-glucose (99.5%) and Span 80 were obtained from Sigma-Aldrich (Poznan, Poland). Sulphuric acid and isopropanol (99%) were purchased from Chempur (Piekary Slaskie, Poland). Acetonitrile (≥99%), methanol (≥99%), calcium chloride (99%) and phosphorus pentoxide (98%) were obtained from VWR Chemicals (Gdansk, Poland). Barium sulfate (95%) and gelatin (95%) were purchased from WarChem (Warsaw, Poland). Sodium alginate from brown algae was obtained from Biomus (Lublin, Poland). Sunflower oil (food grade) was bought from Bunge Polska (Kruszwica, Poland).

### 3.2. Synthesis

Halogenoesters (decyl chloroacetate 10Cl, dodecyl chloroacetate 12Cl, dodecyl bromoacetate 12Br) were obtained from alcohols: 1–decanol or 1-dodecanol with appropriate amount of halogenoacetic acid under acidic conditions (catalytic amount of sulphuric acid) according to of the described esterification reaction procedure [91].

In all gemini surfactants synthesis, 1 equivalent of the appropriate diamine (TMPDA for 10(E)-3-10(E) Cl and 12-3-12; BDMAEE for 10(E)-O-10(E) Cl, 12-O-12, 12(E)-O-12(E) Cl and 12(E)-O-12(E) Br) was mixed with 2 equivalents of halogenoester (10Cl for 10(E)-3-10(E) Cl and 10(E)-O-10(E) Cl; 12Cl for 12(E)-O-12(E) Cl; 12Br for 12(E)-O-12(E) Br), or bromododecane for 12-3-12 and 12-O-12. The reactions were carried out without a solvent, at room temperature, by stirring using a magnetic stirrer until the reaction mixture solidified (max 6 h). The crude products were crystallized from a mixture of acetonitrile:methanol in the volume ratio of 10:1 and dried in an incubator (60 °C) and over P_4_O_10_ in a vacuum desiccator.

### 3.3. Analytical Methods

The melting point (mp) of the products was measured on Stuart SMP30 (Staffordshire, UK) by using a one-sealed side capillary. Measurement accuracy is up to 1 °C.

The NMR spectra for the synthesized compounds were obtained by using a Varian model VNMR-S 400 MHz (Oxford, UK), operating at 403 and 101 MHz for ^1^H and ^13^C, respectively, by using the software, VNMR VERSION 2.3 REVISION A (Varian, Oxford, UK). The ^13^C and ^1^H chemical shifts were measured in CDCl_3_ with TMS as an internal standard.

The FTIR spectra of all gemini surfactants were recorded with a Thermo Scientific Nicolet iS5 FT-IR spectrometer (Waltham, MA, USA). The samples were in a solid phase and prepared as tablets with potassium bromide, whereas the FTIR analysis of alginate and gelatin capsules was performed using a JASCO FT/IR-4600 spectrometer (JASCO, Tokyo, Japan). The samples were tested in the form of solids with a suitable powder attachment (Attenuated Total Reflectance-ATR).

The elemental analysis (EA) measurements were carried out on a FLASH 2000 elemental analyzer (Thermo Fisher Scientific, Warsaw, Poland) with a thermal conductivity detector.

Digital photos were taken with Levenhuk DTX 50 Digital Microscope (Tampa, FL, USA).

Scanning electron microscopy (SEM) studies were performed using a high-resolution environmental electron scanning microscope Quanta FEG 250 FEI (Quanta, Eindhoven, The Netherlands) in low vacuum conditions at a pressure of 70 Pa and a beam accelerating voltage of 10 kV.

### 3.4. Preparation of Alginate Capsules

To fabricate gel capsules of gemini surfactants, an excess amount of suitable gemini surfactant (10(E)-3-10(E) Cl, 10(E)-O-10(E) Cl, 12-3-12, 12-O-12) higher than its critical micelle concentration was added to a mixture 50 mL of sodium alginate (0.6 g) with vigorous stirring at 70 °C for 20 min until reaching homogeneity. After that, the synthesized homogeneous viscous mixture was injected into a calcium chloride solution (2% wt) to form the capsules. Subsequently, the capsules were filtered from the solution and kept in incubator (60 °C) to dry. Empty capsules without gemini surfactant were prepared in an analogous manner.

To prepare more durable capsules of gemini surfactant 12-O-12, an excess amount of was added to a mixture 50 mL of sodium alginate (0.6 g) and barium sulfate (1.2 g) with vigorous stirring at 70 °C for 20 min until reaching homogeneity. After that, the homogeneous viscous mixture was injected into calcium chloride solution (2% wt) to form the capsules. Subsequently, the capsules were collected from the solution and kept in incubator (60 °C) to dry. Empty capsules without gemini surfactant were prepared in an analogous manner.

### 3.5. Preparation of Gelatin Capsules

In total, 1.5 g of α-D-glucose was added to the aqueous gelatin solution (15%, 15 mL, 80 °C) and allowed to react for 5 min. Then, the individual gemini surfactants (about 0.1 g) were added to the aqueous solution containing gelatin and sugar. The resulting mixture was carefully added to sunflower oil (150 mL, 80 °C) containing Span 80 (1.5 mL). The dispersion was mixed with a magnetic stirrer to obtain an emulsion. Quick cooling to 5 °C with an ice bath hardens the gelatin droplets. After 15 min, isopropanol (15 × 4 mL, 5 °C) was added at regular intervals to dehydrate the droplets. Stirring was continued for 2 h, followed by filtration. The microspheres were washed with isopropanol to remove any adhered oil. The microspheres prepared in this way were allowed to dry in a desiccator over P_2_O_5_. Empty capsules without gemini surfactant were prepared in an analogous manner.

## 4. Conclusions

Ensuring environmental safety and the user’s safety requires the use of necessary chemical compounds with the highest activity at the lowest concentrations. Moreover, their operations should be limited to a specific place and activated by a specific stimulus, such as pH, pressure, water etc. Gemini surfactants are modern compounds with a very high biological activity and high anticorrosion efficacy. Immobilization of these compounds in alginate and gelatin microcapsules allow you to introduce them to a specific carrier and obtain smart materials. In this work, we presented effective encapsulated methods of the synthesized gemini surfactants in alginate and gelatin microcapsules. The encapsulation efficiency and the capsule morphology were confirmed by physicochemical tests. The resulting microcapsules meet the smart materials criteria and are promising candidates as corrosion inhibitors and antimicrobials.

## Data Availability

The data presented in this study are available in the article.

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
