# Peer review of "Fabrication of Encapsulated Gemini Surfactants"

_molecules, 2022, doi:10.3390/molecules27196664_

Round 1

Reviewer 1 Report

Title - Fabrication of encapsulated gemini surfactants

Journal name – Molecules

This is a clear, concise and well written manuscript. The work done by Bogumił Brycki at al. is truly commendable.

Corrections

Introduction section is too general. Authors are advised to revise it.

Authors are advised to explain the synthesis process and how they purify the surfactants.

How do you find the exact diameter of wet and dry capsule? (line no. 182)

What is responsible interaction between BaSO4 and capsules for its stability?

When BaSO4 is coated on the microcapsules, why the size is not changing?

Authors are advised to perform EDX analysis for the confirmation of gemini surfactants within the capsules.

What is the application for such encapsulated gemini surfactants? Authors are advised to explore at least one.

Minor Comments

 Improve quality of structures and FTIR spectra

 Calcium hloride – Calcium chloride (line no. 158)

There are several Spelling mistakes in the manuscript. Authors are advised to thoroughly checked the manuscript.

Reviewer 2 Report

The manuscript molecules-1879563 "Fabrication of encapsulated gemini surfactants" describes efficient methods of preparation of encapsulated gemini surfactants in form of alginate and gelatin capsules.

I have the following comments and suggestions for the authors:

1)      There is a lot of self-citation for Bogumił Brycki (18%). It must be corrected.

2)      The authors described the preparation of Gemini surfactants and stayed that the reactions proceeds according to SN2 mechanism. How did the authors confirm this mechanism?

3)      Figures 2, 3 and 9 have low quality. It must be corrected.

4)      The captions for the bands in Figure 9 are hard to distinguish. It must be corrected.

5)      I believe that “Figure 2” would be more correctly called “Scheme 1”. It is also necessary to sign yields for each compounds.

6)      The authors characterized all synthesized compounds in part “2.2 Analysis of Gemini Surfactants” only by 1H and 13C NMR and IR spectroscopy. This is not enough. The authors must to add the mass spectra of the obtained compounds.

7)      –line 156 – “hloride” should be changed to “chloride”.

8)      How did the authors can explain that effectiveness of encapsulation by gelatin capsules (60%) much lower than by alginate capsules (90%)?

9)      –line 265 – “a” should be changed to “an”. “glatin” should be changed to “ gelatin”.

10)  –line 270 – “geimini” should be changed “gemini”.

11)  How did the authors can explain the influence of the type of active substance on the size of the capsules? What is it connected with?

Round 2

Reviewer 1 Report

The authors have responded to all the queries raised by the reviewer. The manuscript can be accepted in its present form. 

Author Response

Dear Reviewer,

Thank you very much for reading and commenting on the manuscript.

Reviewer 2 Report

The authors must to specify yields for all the compounds on Scheme 1.

Author Response

Dear Reviewer,

Thank you very much for reading and commenting on the manuscript.

Yields were specified for all the compounds in Supplementary Materials.